# Derivation and Validation of a Predictive Score for Respiratory Failure Worsening Leading to Secondary Intubation in COVID-19: The CERES Score

**DOI:** 10.3390/jcm11082172

**Published:** 2022-04-13

**Authors:** Alexandre Gaudet, Benoit Ghozlan, Annabelle Dupont, Erika Parmentier-Decrucq, Mickael Rosa, Emmanuelle Jeanpierre, Constance Bayon, Anne Tsicopoulos, Thibault Duburcq, Sophie Susen, Julien Poissy

**Affiliations:** 1Critical Care Center, Department of Intensive Care Medicine, CHU Lille, F-59000 Lille, France; benoit.ghozlan@chu-lille.fr (B.G.); erika.parmentier@chu-lille.fr (E.P.-D.); constance.bayon@gmail.com (C.B.); thibault.duburcq@chu-lille.fr (T.D.); julien.poissy@chu-lille.fr (J.P.); 2Univ. Lille, CNRS, Inserm, CHU Lille, Institut Pasteur de Lille, U1019-UMR9017-CIIL-Centre d’Infection et d’Immunité de Lille, F-59000 Lille, France; anne.tsicopoulos@pasteur-lille.fr; 3Univ. Lille, Inserm, CHU Lille, Institut Pasteur de Lille, U1011-EGID, F-59000 Lille, France; annabelle.dupont@chu-lille.fr (A.D.); mickael.rosa@univ-lille.fr (M.R.); emmanuelle.jeanpierre@chu-lille.fr (E.J.); sophie.susen@chu-lille.fr (S.S.); 4Univ. Lille, Inserm U1285, CHU Lille, CNRS, UMR 8576, UGSF, Unité de Glycobiologie Structurale et Fonctionnelle, F-59000 Lille, France

**Keywords:** COVID-19, late intubation, prediction, score, endothelial

## Abstract

Predictive scores assessing the risk of respiratory failure in COVID-19 mostly focused on the prediction of early intubation. A combined assessment of clinical parameters and biomarkers of endotheliopathy could allow to predict late worsening of acute respiratory failure (ARF), subsequently warranting intubation in COVID-19. Retrospective single-center derivation (*n* = 92 subjects) and validation cohorts (*n* = 59 subjects), including severe COVID-19 patients with non-invasive respiratory support, were assessed for at least 48 h following intensive care unit (ICU) admission. We used stepwise regression to construct the COVID endothelial and respiratory failure (CERES) score in a derivation cohort, and secondly assessed its accuracy for the prediction of late ARF worsening, requiring intubation within 15 days following ICU admission in an independent validation cohort. Platelet count, fraction of inspired oxygen, and endocan measured on ICU admission were identified as the top three predictive variables for late ARF worsening and subsequently included in the CERES score. The area under the ROC curve of the CERES score to predict late ARF worsening was calculated in the derivation and validation cohorts at 0.834 and 0.780, respectively. The CERES score is a simple tool with good performances to predict respiratory failure worsening, leading to secondary intubation, in COVID-19 patients.

## 1. Introduction

COVID-19 can be responsible for severe hypoxemia, which may warrant the use of invasive mechanical ventilation, occurring beyond 48 h after intensive care unit (ICU) admission in half of the patients [1,2].

Noninvasive respiratory strategies such as continuous positive airway pressure (CPAP) and high-flow nasal oxygen (HFNO) have been proposed to avoid invasive mechanical ventilation in patients with COVID-19-related acute hypoxemic respiratory failure [3]. Recent data report that more than half of the patients admitted in the ICU for severe COVID-19 during the Omicron wave in the US exclusively underwent non-invasive strategies [4]. However, it has been shown that delayed intubation is associated with increased mortality in critically ill patients on HFNO therapy [5], especially in the setting of COVID-19 pneumonia [6,7]. This phenomenon likely results from intense spontaneous inspiratory efforts associated with high transpulmonary pressures, contributing to the development of self-induced lung injury [7]. Therefore, the identification of patients at high risk of progression to respiratory failure and requiring intubation is of major interest for the management of severe COVID-19, as this subgroup of patients could benefit from early intubation.

Among the means to predict the risk of failure of non-invasive strategies in severe COVID-19, the ROX index, based on the clinical assessment of respiratory rate and pulse oxygen saturation, has been widely investigated. However, studies which evaluated this score mainly assessed its performances for the prediction of early intubation [8]. Consequently, the search for tools to predict the late failure of non-invasive strategies in patients with severe COVID-19 remains of major interest.

Previous studies have focused on the profile of biomarkers of endotheliopathy in severe COVID-19. Among these, some markers were found to be associated with a poor outcome [9,10]. Nevertheless, the prognostic values of these markers taken individually remain insufficient to determine with optimal accuracy the risk of progression to unfavorable respiratory outcome. Furthermore, no study to date has evaluated the value of phenotypic categorization based on the combined assessment of these markers. 

Subsequently, the aim of our work was to derive and validate the COVID-related endotheliopathy and respiratory failure (CERES) score, based on routinely available clinical variables and endotheliopathy biomarkers for the prediction of late worsening of acute respiratory failure (ARF), subsequently requiring intubation in severe COVID-19.

## 2. Materials and Methods

### 2.1. Study Design and Patients

This study was conducted in a 74-bed mixed ICU (Critical Care Centre, CHU of Lille, Lille, France). We retrospectively included patients who underwent blood samplings between 7 August 2020 and 30 April 2021 in the setting of the PREDICT study (Clinicaltrial registration: NCT04327180) and who met the following criteria: age 18 or over, a positive SARS-CoV-2 real-time polymerase chain reaction, patients undergoing HFNO and/or non-invasive ventilation (NIV) and/or CPAP within the first 48 h following ICU admission, and treatment by intravenous corticosteroids at a dose regimen ranging from 6 to 20 mg/day of equivalent Dexamethasone on ICU admission. The exclusion criteria were a need for tracheal intubation within 48 h following ICU admission and a limitation of therapeutic benefit on tracheal intubation. Patients with sufficient blood samples available to perform a complete assessment of all biomarkers explored in our study were included in the derivation cohort, while patients with insufficient blood samples available were included in the validation cohort.

### 2.2. Ethics Statement

This research was examined and approved by the Institutional Review Board of Saint-Louis Hospital, Paris, France (Comité de Protection des Personnes Ile de France IV, approval number: 2020/30). Written informed consent was obtained from all the patients or their relatives.

### 2.3. Data Collection

Patient demographic and clinical characteristics, severity scores, and comorbidities were recorded at baseline for all patients. Microbiological tests performed within 48 h of admission were also recorded. Relevant tests included blood and sputum cultures, bronchoalveolar lavage fluids, multiplex PCRs on sputum, and urinary antigen tests for *Legionella pneumophila* and *Streptococcus pneumoniae*. Further, data about clinical outcomes during the stay in the ICU were obtained for all patients.

### 2.4. Laboratory Testing

Blood was collected at ICU admission for each patient on 0.109 mol/L trisodium citrate tube (BD Vacutainer, BD Diagnostics, Sparks, MD, USA). All analyses were performed on platelet-poor plasma obtained after a double centrifugation at 2500× *g* for 15 min at room temperature.

The following biomarkers were measured in the derivation and validation cohorts: Prothrombin time and fibrinogen were measured on an STA-R Max analyzer (Diagnostica Stago, Asnières-sur-Seine, France) using STA^®^ Neoplastin^®^ R (Diagnostica Stago) and STA^®^Liquid-Fib (Diagnostica Stago); D-dimers levels were measured in µg/mL (fibrinogen equivalent units) using an immunoturbidimetric latex-particle assay (Liatest^®^ DDI-Plus, Diagnostica Stago) on the STA-R Max analyzer (Diagnostica Stago); and Von Willebrand factor antigen (VWF:Ag) was measured using an immunoturbidimetric assay (LIAPHEN^®^ VWF:Ag, HYPHEN BioMed, Andresy, France) on a CS 2400 analyzer (Sysmex, Kobe, Japan). Endocan was measured using the JDIYEK^®^ ELISA Kit (Biothelis, Lille, France). Other laboratory blood tests including a complete blood count, CRP, procalcitonin (PCT), liver transaminases, bilirubin, and creatinine were measured by standard methods as part of the patient’s care in the Biology and Pathology Center of Lille University Hospital.

### 2.5. Definitions

Criteria from the 11-point World Health Organization (WHO) clinical progression scale [11] were used for the definition of respiratory failure.

The end-point variable of our prediction model was the occurrence of late ARF worsening defined as a progression beyond 48 h following ICU admission on WHO clinical progression scale score from 6 at admission to ≥7 within 15 days following ICU admission (D15), corresponding to patients who underwent invasive mechanical ventilation or who did not survive on the day of outcome assessment at D15. The follow-up period was limited to 15 days to minimize the contribution of long-term ICU complications unspecific to COVID-19 as confounding events in the outcome assessment.

### 2.6. Statistical Analysis

Categorical variables were expressed as numbers (percentages) and compared using Chi-square test or Fisher’s exact test, as appropriate. Normality of distribution of continuous variables was checked graphically and by using the Shapiro–Wilk test. Normally distributed continuous variables are presented as means (SD) and compared using Student’s *t*-test.

We performed a missing value imputation for continuous variables in the derivation cohort using the predictive mean-matching method with the *mice* function in R (*mice* R package) [12]. No missing data were reported for outcomes variables, neither for categorical variables, with the exception of pulmonary embolism on CT-scan on ICU admission (missing because no CT-scan with injection was performed on ICU admission for 31/92 and 22/59 subjects in the derivation and validation cohorts, respectively) and predominant findings on CT-scan (missing because no CT-scan was performed on ICU admission for 20/92 and 12/59 subjects in the derivation and validation cohorts, respectively). Counts of missing data for continuous variables are shown in Appendix A. Variables with missing data exceeding 30% of total count were excluded from the analysis.

We included the following exposure variables in a prediction model: age, gender, diabetes, chronic respiratory failure, chronic obstructive pulmonary disease (COPD), chronic heart failure, cirrhosis Child B or C, end stage kidney disease, immunosuppression, pulmonary embolism on ICU admission, body mass index (BMI), BMI > 30 kg/m^2^, simplified acute physiology score 2 (SAPS 2), sequential organ failure assessment (SOFA), computed tomography (CT)-scan extension, fraction of inspired oxygen (FiO_2_), VWF:Ag, Angiopoietin 2, VEGF, syndecan-1, endocan, suPAR, PAI-1, TFPI, CRP, PCT, LDH, ALAT, ASAT, total bilirubin, creatinine, ferritin, TQ ratio, fibrinogen, D-dimers and hemoglobin levels, leucocytes, neutrophils, lymphocytes, and platelets count.

Immunosuppression included solid organ transplantation, hematopoietic stem cell transplantation in the last 6 months, corticosteroid therapy at a dosage higher than or equivalent to prednisone 1 mg/kg/day for more than 3 months, uncontrolled human immunodeficiency virus infection (<200 CD4/mm^3^), and active solid malignant tumor or hematologic malignancy in the last 6 months. FiO_2_ was defined as the highest FiO_2_ value recorded at any time within 48 h following ICU admission. CT-scan extension was defined at the lung parenchymal extension of pneumonia on CT-scan performed on the day of admission in the ICU.

We used univariate and multivariate logistic regression in the derivation dataset to estimate the coefficients of each potential predictor of late ARF worsening on D15. We selected the predictor variables that contribute the most to the model using the stepwise regression method. We therefore fitted the final model as the one with the smallest Akaike information criterion, allowing to maximize the predictive performance, while limiting model complexity. We assessed the predictive performance of the model in the derivation and validation cohorts by calculating its area under receiver operating characteristics (AUROC) to predict late ARF worsening at D15. Sensitivity and specificity were calculated for different scores. We finally computed Kaplan–Meier curves for late ARF worsening at D15 between groups with predefined CERES values and compared the probability of late ARF worsening between groups using the Logrank test.

All statistical tests were two-tailed, and *p*-values < 0.05 were considered statistically significant. The statistical analyses were performed using R version 4.1.2 (R foundation for statistical computing, Vienna, Austria).

## 3. Results

### 3.1. Patient Characteristics

Characteristics of the patients included in our study are shown in Table 1. The derivation cohort included 92 patients, of which 29 met the criteria for late ARF worsening on D15. The validation cohort included 59 patients, of which 14 met the criteria for late ARF worsening on D15. No patients enrolled in this study underwent veno-venous ECMO during their stay in ICU.

In the derivation cohort, subjects with late ARF worsening on D15 were older (mean (SD): 69 (8) years in the late ARF-worsening group vs. 63 (13) years in the non-late ARF-worsening group, *p* = 0.02), had higher FiO_2_ values (mean (SD): 88 (16) in the late ARF-worsening group vs. 77 (20) in the non-late ARF-worsening group, *p* = 0.01), higher mortality in ICU (N = 22 (76%) in the late ARF-worsening group vs. 1 (2%) in the non-late ARF-worsening group, *p* < 10^−3^), and longer ICU stay (mean (SD): 21 (15) days in the late ARF-worsening group vs. 9.5 (12) days in the non-late ARF-worsening group, *p* < 10^−3^).

In the validation cohort, subjects with late ARF worsening had a higher frequency of COPD (N = 6 (43%) in the late ARF-worsening group vs. 5 (11%) in the non-late ARF-worsening group, *p* = 0.01), higher FiO_2_ values (mean (SD): 84 (16) in the late ARF-worsening group vs. 70 (20) in the non-late ARF-worsening group, *p* = 0.01), higher mortality in ICU (N = 11 (79%) in the late ARF-worsening group vs. 1 (2%) in the non-late ARF-worsening group, *p* < 10^−3^) and longer ICU stay (mean (SD): 27 (17) days in the late ARF-worsening group vs. 8 (5) days in the non-late ARF-worsening group, *p* < 10^−3^).

### 3.2. Biomarkers on ICU Admission

Results of biomarker levels assessed on ICU admission are shown in Table 2.

In the derivation cohort, patients with late ARF worsening had higher levels of endocan (mean (SD): 9.13 (15.34) ng/mL in the late ARF-worsening group vs. 3.39 (3.08) ng/mL in the non-late ARF-worsening group, *p* < 10^−2^), suPAR (mean (SD): 7.52 (4.01) ng/mL in the late ARF-worsening group vs. 6.22 (2.21) ng/mL in the non-late ARF-worsening group, *p* = 0.048), PCT (mean (SD): 6.45 (21.88) ng/mL in the late ARF-worsening group vs. 0.57 (0.74) ng/mL in the non-late ARF-worsening group, *p* = 0.03), D-dimers (mean (SD): 6.85 (15.53) µg/mL in the late ARF-worsening group vs. 2.13 (3.39) µg/mL in the non-late ARF-worsening group, *p* = 0.02), and lower platelets levels (mean (SD): 200 (65) G/L in the late ARF-worsening group vs. 274 (109) G/L in the non-late ARF worsening group, *p* < 10^−2^).

In the validation cohort, patients from the late ARF-worsening group had lower fibrinogen levels (mean (SD): 6.05 (1.62) g/L in the late ARF-worsening group vs. 7.06 (1.65) g/L in the non-late ARF-worsening group, *p* = 0.049) and fewer leucocytes (mean (SD): 5.57 (2.28) G/L in the late ARF-worsening group vs. 7.84 (3.89) G/L in the non-late ARF-worsening group, *p* = 0.04).

### 3.3. Derivation and Validation of the CERES Score

According to the stepwise regression performed in our derivation cohort, the platelets level was identified as the variable exhibiting the strongest association with late ARF worsening on D15 (per additional G/L, OR (95% CI): 0.99 (0.98–0.99)), followed by FiO_2_ (per additional %, OR (95% CI): 1.05 (1.02–1.09)) and endocan (per additional ng/mL, OR (95% CI): 1.13 (1.04–1.31)) (Figure 1).

We then computed the AUROC to predict late ARF on D15 for: (1) the CERES score, calculated as −1.04 × platelets (G/L) + 5.12 × FiO_2_ (%) + 10.4 × endocan (ng/mL); (2) the top two predicting variables-based model, calculated as: −1.09 × platelets (G/L) + 4.11 × FiO_2_ (%); (3) platelets, identified as the top predicting variable in our model. The AUROC for prediction of late ARF worsening on D15 in the derivation and validation cohorts was calculated, respectively, at 0.834 (*p* < 10^−3^) and 0.780 (*p* < 10^−2^) for the CERES score, 0.796 (*p* < 10^−3^) and 0.769 (*p* < 10^−2^) for the top two predicting variables-based model, and 0.719 (*p* < 10^−3^) and 0.666 (*p* = 0.06) for platelets (Figure 2a). According to the Youden index, the best cut-offs for the detection of late ARF worsening on D15 were found respectively for CERES values ≥ 216 in the derivation cohort (sensitivity = 0.9, specificity = 0.7) and 232 in the validation cohort (sensitivity = 0.71, specificity = 0.73). A CERES value ≥ 140 showed the best sensitivity to predict late ARF worsening on D15, observed respectively at 1 and 0.93 in the derivation and validation cohorts, while specificity was found at 0.48 and 0.44, respectively. Further, the best specificity for the prediction of late ARF worsening on D15 was observed for a CERES value ≥ 333, being then respectively found at 0.9 and 0.93 in the derivation and validation cohorts, while sensitivity was measured at 0.45 and 0.5 with this cut-off (Figure 2b).

Survival analysis revealed in the derivation and validations cohorts a higher probability of late ARF on D15 in the case of CERES ≥ 140 vs. CERES < 140 (*p* < 10^−3^ in both cohorts) (Figure 3), and in the case of CERES ≥ 333 vs. CERES < 333 (*p* < 10^−3^ and *p* = 0.012, respectively) (Figure 4).

## 4. Discussion

Non-invasive techniques remain frequently used as respiratory support for patients admitted to the ICU for COVID-19 [3,4]. Nevertheless, the risk of failure of non-invasive methods remains high [4], then being associated with an increased severity of COVID-induced lung injury in the case of late intubation, and consequently, with a poor prognosis for patients [6,7].

The main purpose of this study was to develop a score to early identify, at ICU admission, patients at high risk of late failure of the non-invasive respiratory management strategy in severe COVID-19. We hereby report the results from derivation and validation of the CERES score, based on the assessment of three variables: platelets, FiO_2_, and endocan, to predict the risk of late respiratory failure requiring intubation in critically ill COVID patients.

Our findings regarding the factors associated with poor outcomes in COVID-19 are consistent with previously published data from the literature. Indeed, the association between thrombocytopenia and severity of illness in COVID-19 was consistently reported early in the pandemic [13]. This association may result from the endothelial injury with endothelial cell membrane disruption occurring in severe COVID-19 [9,14]. On the other hand, a positive correlation has been reported between the severity of COVID-19 and circulating levels of endocan [10,15], an endothelial-cell borne biomarker secreted in the deep lung under inflammatory conditions [16]. These results are consistent with the view that endocan is a marker of lung injury, therefore being elevated in conditions of inflammation-induced acute respiratory failure [17,18,19]. Furthermore, FiO_2_ has been widely used as a key component of predictive models for COVID-19-related outcomes, such as the ROX index [8,20,21,22,23]. Nonetheless, this clinical score has been mostly described as a good predictor of an imminent need for intubation, mostly occurring within less than 24 h [8], contrasting with the setting of our study, which explored the accuracy of the CERES score to predict the need for invasive mechanical ventilation beyond 48 h.

In addition to the ROX index, other tools have been developed to predict the risk of intubation in COVID-19. Thus, Liu, et al. developed a nomogram to predict the failure of noninvasive respiratory strategies with an AUROC greater than 0.8. However, this study did not specifically address the question of late intubation. Moreover, the use of the nomogram described by the authors may be complex in daily practice, in particular because of a higher number of variables than in the CERES score [24]. In addition, several authors have reported the high performance of machine-learning algorithms to predict the need for invasive mechanical ventilation in patients with COVID-19. Nevertheless, the use of these tools remains particularly complex and difficult to apply at the patient’s bedside [25,26].

Two distinct clinical applications may be derived from our results. First, we found that a CERES value ≥ 140 has a very high sensitivity for the detection of patients who will eventually require invasive respiratory support. Accordingly, the non-invasive strategy was successful in nearly all patients with a CERES score < 140. Based on our results, we could therefore propose to maintain non-invasive strategies in patients with a CERES score < 140. This latter point seems of major importance, as growing evidence supports the hypothesis that delayed intubation and late failure of non-invasive strategies are associated with an increase in mortality and worse pulmonary sequelae [7,27]. In the same view, a CERES score ≥ 333, which detected the further need for invasive support with high specificity, could be used to propose earlier intubation in patients with the highest risk of failure of non-invasive strategies.

Our study has several limitations. Firstly, some variables were collected retrospectively, thus resulting in missing data for several exposition factors analyzed in our prediction model. However, counts of missing data remained limited for most variables, allowing to perform data imputation to conduct our analysis. Importantly, missing data in our derivation cohort concerned 34% of patients for pulmonary embolism and 29% of patients for lung parenchymal extension on chest CT. This could have impacted the integration in our score of these two variables, which have been described as predictive of poor prognosis in COVID-19 [28,29,30]. Secondly, the limited sample sizes used in our study did not allow subgroup analysis of the performances of the CERES score while these could have been of interest. In addition, our sample sizes may question the extrapolability of our findings. Nevertheless, the results observed in our derivation cohort were almost completely reproduced in our validation cohort, especially regarding sensitivities and specificities associated with CERES values at 140 and 333, thus strengthening the reliability of our findings. Thirdly, the single-center design of our study appears as a limitation to its broader applicability. However, patients’ characteristics at ICU admission in our derivation and validation cohorts were similar to those reported in wider multi-center cohorts of patients with severe COVID-19 undergoing non-invasive respiratory support [3]. Fourthly, we found higher PCT values in subjects with late ARF worsening in the derivation cohort, raising the question of a higher frequency of bacterial coinfections in these patients. However, both groups had similar rates of microbiological confirmation. In addition, PCT values were not found to be different between groups in the validation cohort, with little influence on the performances of the CERES score, which appeared to be preserved in this cohort. Fifthly, nearly half of the patients from our study had a CERES score in the “grey zone”. Indeed, CERES values between the highly sensitive and specific boundaries, respectively, set at 140 and 330, concerned 45/92 subjects in the derivation cohort and 28/59 subjects in the validation cohort. Subsequently, our model seems too inaccurate to be used in the decision process of whether to perform intubation or not in those patients. Conversely, the CERES score may be used to guide the decision of intubation in more than half of the patients, thus appearing as a helpful means to accurately identify the most appropriate strategy in terms of respiratory support. Finally, subjects included in our study were recruited in a time frame when the original strain and the alpha variant of COVID-19 accounted for most cases of the disease. Further prospective investigations in larger cohorts are warranted to confirm our results. 

## 5. Conclusions

In patients with severe COVID-19 undergoing non-invasive respiratory support, CERES values < 140 and ≥ 333 appear as accurate predictors of late ARF worsening with subsequent need for intubation. Accordingly, our results suggest that these cut-offs could be used, at ICU admission, to drive the decision of intubation in these patients.

## Figures and Tables

**Figure 1 jcm-11-02172-f001:**
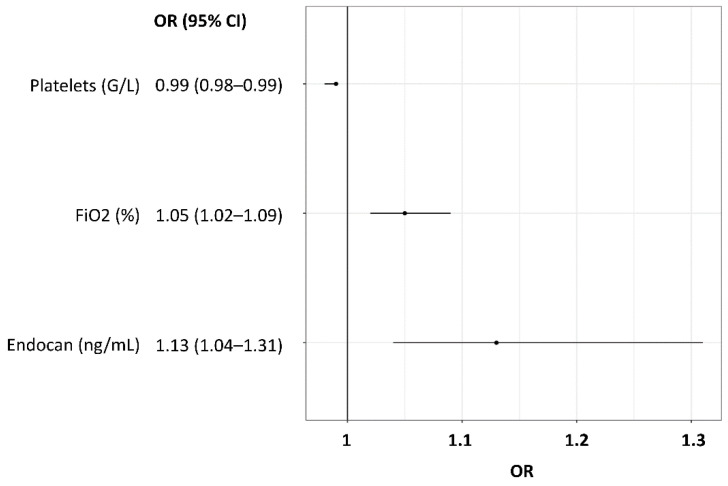
Odds ratios for the prediction of late ARF worsening at D15 by multivariate logistic regression in the derivation cohort. Results are shown as OR with 95% CI for the top three predictive variables, included in the CERES score. ARF—acute respiratory failure, D15—day 15 following ICU admission, FiO_2_—fraction of inspired oxygen, OR—odds ratio.

**Figure 2 jcm-11-02172-f002:**
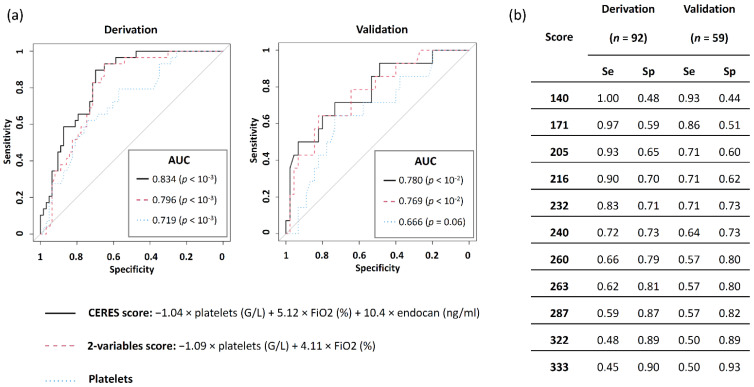
ROC curves for the prediction of late ARF worsening at D15 in the derivation and validation cohorts. (**a**) ROC curves were computerized in derivation and validation cohorts for the CERES score (black plain line) for the top two predictive variables-based model including platelets and FiO_2_ (red dashed line), and for platelets (blue dotted line). (**b**) Values of sensitivity and specificity for the CERES score. ARF—acute respiratory failure, AUC—area under the curve, D15—day 15 following ICU admission, FiO_2_—fraction of inspired oxygen, Se—sensitivity, Sp—specificity.

**Figure 3 jcm-11-02172-f003:**
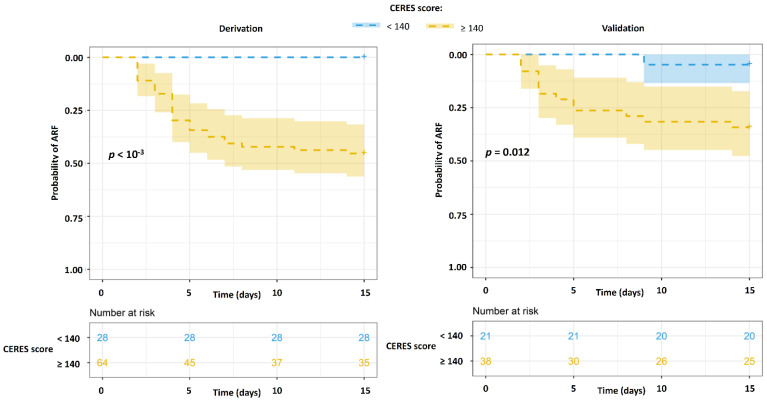
Kaplan–Meier curves of late ARF worsening at D15 in the derivation and validation cohorts according to the high-sensitivity CERES cut-off. Probabilities of late worsening of ARF on D15 in subjects with CERES ≥ 140 (yellow) vs. subjects with CERES < 140 (blue). Probabilities of survival measured in our cohorts are displayed by dashed lines. Colored areas represent 95% CI. ARF—acute respiratory failure, D15—day 15 following ICU admission.

**Figure 4 jcm-11-02172-f004:**
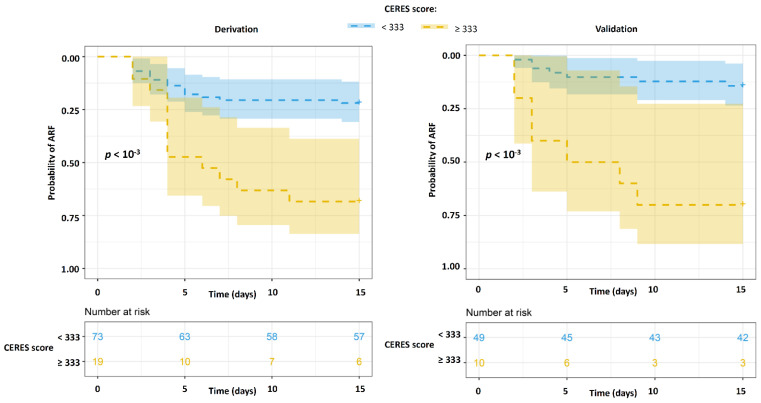
Kaplan–Meier curves of late ARF worsening at D15 in the derivation and validation cohorts according to the high-specificity CERES cut-off. Probabilities of late worsening of ARF on D15 in subjects with CERES ≥ 333 (yellow) vs. subjects with CERES < 333 (blue). Probabilities of survival measured in our cohorts are displayed by dashed lines. Colored areas represent 95% CI. ARF—acute respiratory failure, D15—day 15 following ICU admission.

**Table 1 jcm-11-02172-t001:** Patients’ characteristics.

Variable	Derivation Cohort			Validation Cohort		
	ARF Worsening at D15		*p*	ARF Worsening at D15		*p*
	No (*n* = 63)	Yes (*n* = 29)		No (*n* = 45)	Yes (*n* = 14)	
**Demographics**						
Age, years	**63 (13)**	**69 (8)**	**0.02**	63 (14)	64 (13)	0.73
BMI, kg/m^2^	31 (6)	29 (5)	0.26	31 (7)	32 (6)	0.65
Gender, female	16 (25)	5 (17)	0.55	13 (29)	2 (14)	0.48
No comorbidities ^α^	17 (27)	10 (34)	0.46	10 (22)	4 (29)	0.72
BMI > 30	29 (46)	9 (31)	0.17	21 (47)	7 (50)	0.83
Diabetes	20 (32)	9 (31)	1	14 (31)	4 (29)	1
Chronic respiratory failure	4 (6)	6 (21)	0.09	0 (0)	1 (7)	0.24
COPD	10 (16)	6 (21)	0.79	**5 (11)**	**6 (43)**	**0.01**
Chronic heart failure	6 (9)	4 (14)	0.8	5 (11)	3 (21)	0.38
Cirrhosis Child B or C	1 (2)	0 (0)	1	0 (0)	0 (0)	1
End stage kidney disease ^β^	4 (6)	4 (14)	0.44	3 (7)	2 (14)	0.58
Immunocompromised ^γ^	4 (6)	6 (21)	0.09	6 (13)	2 (14)	1
**Characteristics of disease on ICU admission**						
SAPS2	35 (8)	38 (8)	0.2	32 (11)	37 (7)	0.14
SOFA	2.5 (0.9)	2.8 (1.5)	0.2	2.8 (1.4)	2.9 (1.3)	0.84
FiO_2_, %	**77 (20)**	**88 (16)**	**0.01**	**70 (20)**	**84 (16)**	**0.01**
CT-scan extension, %	48 (19)	45 (20)	0.62	47 (21)	60 (21)	0.08
Predominant findings on CT-scanGround-glass opacities	39 (62)	17 (59)	0.76	26 (58)	10 (71)	0.55
Consolidation	13 (21)	3 (10)	0.36	8 (18)	3 (21)	0.71
Pulmonary embolism	10 (16)	6 (21)	0.79	3 (7)	1 (7)	1
Purulent sputum	8 (13)	2 (7)	0.5	5 (11)	2 (14)	0.67
Microbiologically confirmed bacterial co-infection ^δ^	4 (6)	2 (7)	1	5 (11)	0 (0)	0.33
Use of antibiotics prior to collection of microbiological specimens	20 (32)	10 (34)	0.98	12 (27)	7 (50)	0.12
**Treatments on ICU admission**						
CPAP H0–H48 ^ϕ^	20 (32)	13 (45)	0.33	19 (42)	5 (36)	0.9
NIV H0–H48 ^χ^	31 (49)	21 (72)	0.06	18 (40)	10 (71)	0.08
Prone positioning H0–H48	11 (17)	3 (10)	0.57	9 (20)	2 (14)	1
Antibiotics	43 (68)	20 (69)	1	40 (89)	13 (93)	1
Tocilizumab	2 (3)	4 (14)	0.14	1 (2)	0 (0)	1
Remdesivir	5 (8)	4 (4)	0.62	4 (9)	0 (0)	0.56
**Outcomes**						
ICU mortality	**1 (2)**	**22 (76)**	**<10^−3^**	**1 (2)**	**11 (79)**	**<10^−3^**
ICU length of stay, days	**9 (12)**	**21 (15)**	**<10^−3^**	**8 (5)**	**27 (17)**	**<10^−3^**

Data are presented as number (%) or mean (SD). Numbers are indicated in bold characters in case of *p*-values < 0.05. ARF—acute respiratory failure requiring intubation, D15—day 15 following ICU admission, BMI—body mass index, COPD—chronic obstructive pulmonary disease, CPAP—continuous positive airway pressure, CT—computed tomography, FiO_2_—fraction of inspired oxygen, ICU—intensive care unit, NIV—non-invasive ventilation, SAPS 2—simplified acute physiology score 2, SOFA—sequential organ failure assessment. ^α^ No comorbidities among the following ones: BMI > 30, diabetes, chronic respiratory failure, COPD, chronic heart failure, cirrhosis child B or C, end stage kidney disease, and immunocompromised. ^β^ End-stage kidney disease, defined as an estimated glomerular filtration rate eGFR of <30 mL/minute/1.73 m^2^ or patients undergoing renal replacement therapy. ^γ^ Immunocompromised subjects exhibited at least one of the following conditions: solid organ transplantation, hematopoietic stem cell transplantation in the last 6 months, corticosteroid therapy at a dosage higher than or equivalent to prednisone 1 mg/kg/day for more than 3 months, uncontrolled human immunodeficiency virus infection (<200 CD4/mm^3^), active solid malignant tumor or hematologic malignancy in the last 6 months. ^δ^ Microbiologically confirmed bacterial co-infection defined by the positivity of any microbiological test performed within 48 h of admission, including blood and sputum cultures, bronchoalveolar lavage fluids, multiplex PCRs on sputum, and urinary antigen tests for *Legionella pneumophila* and *Streptococcus pneumoniae*. ^ϕ^ CPAP was defined as the delivery of a preset pressure that is constant during both inhalation and exhalation using either a ventilator with dedicated settings or a Boussignac valve. ^χ^ NIV was defined as the delivery of a predetermined pressure (or volume) during inspiration in addition to providing positive end-expiratory pressure during exhalation. All subjects were treated by corticosteroids on ICU admission, accordingly with the inclusion criteria of our study.

**Table 2 jcm-11-02172-t002:** Biomarkers on ICU admission.

Variable	Derivation Cohort			Validation Cohort		
	ARF Worsening at D15		*p*	ARF Worsening at D15		*p*
	No (*n* = 63)	Yes (*n* = 29)		No (*n* = 45)	Yes (*n* = 14)	
VWF:Ag, %	458 (129)	466 (125)	0.79	422 (101)	418 (120)	0.91
Angiopoietin 2, pg/mL	2311(1312)	3042 (2306)	0.06	--	--	--
VEGF, pg/mL	161 (127)	133 (92)	0.29	--	--	--
Syndecan, ng/mL	209 (239)	297 (515)	0.27	--	--	--
Endocan, ng/mL	**3.39 (3.08)**	**9.13 (15.34)**	**<10^−2^**	4.04 (3.73)	7.93 (17.3)	0.16
suPAR, ng/mL	**6.22 (2.21)**	**7.52 (4.01)**	**0.048**	--	--	--
PAI-1, ng/mL	84.9 (63.7)	81.5 (36.5)	0.79	--	--	--
TFPI, ng/mL	112 (43)	123 (65)	0.33	--	--	--
CRP, mg/L	144 (89)	166 (99)	0.29	155 (92)	128 (74)	0.33
PCT, ng/mL	**0.57 (0.74)**	**6.45 (21.88)**	**0.03**	3.34 (11.21)	1.28 (2.59)	0.52
LDH, UI/L	510 (190)	570 (188)	0.16	507 (153)	640 (459)	0.16
ALAT, UI/L	49.4 (34.1)	45.6 (25.8)	0.6	78.5 (107.2)	83 (111.9)	0.89
ASAT, UI/L	60.8 (31.7)	71.1 (38.3)	0.18	92.8 (110)	147 (220)	0.22
Total bilirubin, mg/L	4.86 (2.17)	5.45 (2.34)	0.24	5.44 (3.09)	6.29 (5.06)	0.45
Creatinine, mg/L	11.6 (17.3)	14.8 (18.1)	0.42	15.2 (20.7)	11.5 (5.2)	0.51
Ferritin, µg/L	1952 (1590)	2153 (1821)	0.59	2246 (2496)	1848 (1159)	0.63
TQ ratio	1.27 (0.69)	1.16 (0.42)	0.45	1.24 (0.40)	1.15 (0.17)	0.42
Fibrinogen, g/L	7.24 (1.58)	6.8 (1.21)	0.19	**7.06 (1.65)**	**6.05 (1.62)**	**0.049**
DDimers, µg/mL	**2.13 (3.39)**	**6.85 (15.53)**	**0.02**	2.41 (1.94)	2.78 (4.41)	0.68
Hemoglobin, g/dL	12.9 (1.6)	12.9 (2.5)	0.86	12.9 (2)	13.1 (2)	0.76
Leucocytes, (G/L)	9.84 (4.3)	9.65 (5.13)	0.86	**7.84 (3.89)**	**5.57 (2.28)**	**0.04**
Neutrophiles, (G/L)	8.52 (3.84)	8.80 (4.52)	0.76	--	--	--
Lymphocytes, (G/L)	0.82 (0.54)	0.99 (1.54)	0.44	--	--	--
Platelets (G/L)	274 (109)	200 (65)	**<10^−2^**	237 (92)	187 (68)	0.07

Data are presented as mean (SD). Numbers are indicated in bold characters in case of *p*-values < 0.05. ARF—acute respiratory failure requiring intubation, D15—day 15 following ICU admission, ICU—intensive care unit, PAI-1—plasminogen activator inhibitor-1, suPAR—soluble urokinase plasminogen activator receptor, TFPI—tissue factor pathway inhibitor, VEGF—vascular endothelial growth factor, VWF:Ag—Von Willebrand factor antigen.

## Data Availability

The datasets used and/or analyzed during the current study are available from the corresponding author on reasonable request.

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
