# Peer review of "Derivation and Validation of a Predictive Score for Respiratory Failure Worsening Leading to Secondary Intubation in COVID-19: The CERES Score"

_jcm, 2022, doi:10.3390/jcm11082172_

Round 1

Reviewer 1 Report

Thank for the opportunity to read your manuscript.

I have minor suggestions to enhance the readability of your work:

  • Line 45, Change "Platelet" to "Platelet count" or something comparable to inform readers you were not referring to platelet function.  
  • Line 66, "patient self-induced" is redundant, self-induced confers the concept.
  • Line 97, Considering using the word "benefit" instead of "effort" in the phrase "a limitation of therapeutic effort on tracheal intubation."
  • Line 318, consider revising to, "... at ICU admission,..."
  • In your discussion,
    • under limitations you don't discuss the impact of the lack of chest CT (with or without angiogram) other than to speak generically about missing data
    • you don't speak to your work being performed in a single medical center as a limitation of its broader applicability.

Reviewer 2 Report

This study aims to develop a CERES score system that could provide early identification of severe COVID-19 patients at high risk of succumbing to failures in non-invasive respiratory management and therapies. A successful scoring system will be crucial in improving the general management of COVID-19 patients presented in the clinical settings. I have several comments on this manuscript:

1) In line with the standard terms used on a global scale and as presented in the title of the manuscript, "COVID-19" should be written in uppercase letters. However, the disease was consistently presented as "Covid-19" across the manuscript.

2) The authors could discuss further on how the CERES score presented in this manuscript fare against the other available scoring systems in managing severe COVID-19 patients.

3) Did the authors observe any difference in the performance of this CERES scoring system between different genders, as men are generally more at risk for severe outcomes following infection with SARS-CoV-2?
